# Relationship between Sugarcane *eIF4E* Gene and Resistance against *Sugarcane Streak Mosaic Virus*

**DOI:** 10.3390/plants12152805

**Published:** 2023-07-28

**Authors:** Hongli Shan, Du Chen, Rongyue Zhang, Xiaoyan Wang, Jie Li, Changmi Wang, Yinhu Li, Yingkun Huang

**Affiliations:** Sugarcane Research Institute, Yunnan Academy of Agricultural Science, Yunnan Key Laboratory of Sugarcane Genetic Improvement, Kaiyuan 661699, China; shhldlw@163.com (H.S.); lbxnyzhfwzx@163.com (D.C.); rongyuezhang@hotmail.com (R.Z.); xiaoyanwang402@sina.com (X.W.); lijie0988@163.com (J.L.); wcmlucky@163.com (C.W.); liyinhu93@163.com (Y.L.)

**Keywords:** eukaryotic translation initiation factor 4E (eIF4E), *sugarcane streak mosaic virus* (*SCSMV*), resistance difference, amino acid variation, non-synonymous mutation

## Abstract

Sugarcane mosaic disease, mainly caused by *Sugarcane streak mosaic virus* (*SCSMV*), has serious adverse effects on the yield and quality of sugarcane. Eukaryotic translation initiation factor 4E (*eIF4E*) is a natural resistance gene in plants. The *eIF4E*-mediated natural recessive resistance results from non-synonymous mutations of the eIF4E protein. In this study, two sugarcane varieties, CP94-1100 and ROC22, were selected for analysis of their differences in resistance to *SCSMV.* Four-base missense mutations in the ORF region of *eIF4E* resulted in different conserved domains. Therefore, the differences in resistance to *SCSMV* are due to the inherent differences in *eIF4E* of the sugarcane varieties. The coding regions of *eIF4E* included 28 SNP loci and no InDel loci, which were affected by negative selection and were relatively conserved. A total of 11 haploids encoded 11 protein sequences. Prediction of the protein spatial structure revealed three non-synonymous mutation sites for amino acids located in the cap pocket of eIF4E; one of these sites existed only in a resistant material (Yuetang 55), whereas the other site existed only in a susceptible material (ROC22), suggesting that these two sites might be related to the resistance to *SCSMV*. The results provide a strong basis for further analysis of the functional role of *eIF4E* in regulating mosaic resistance in sugarcane.

## 1. Introduction

Sugarcane (*Saccharum officinarum* L.) is an important sugar and energy crop in China and the world and is the raw material for 78% of world sugar production. In sugarcane production, sugarcane mosaic disease (SMD) is one of the most common and serious viral diseases, named sugarcane cancer, which seriously affects the yield and quality of sugarcane. *Sugarcane streak mosaic virus* (*SCSMV*), which has been identified as a major virus causing SMD in recent years, belongs to the genus *Poacevirus* in the *Potyviridae* family [1,2,3,4]. The genome of *SCSMV* is composed of a sense single-stranded RNA, about 9.7–10.0 kb in size. Eleven functional proteins are produced from the N-terminal to C-terminal by hydrolase cutting, including Protein 1 (P1), Helper component-proteinase (HC-Pro), Protein 3 (P3), P3N-PIPO (P3 N-terminal fused with Pretty Interesting *Potyviridae* ORF), the first protein of 6 kD (6K1), Cylindrical inclusion, the second protein of 6 kD (6K2), Viral protein genome-linked (VPg), the protease of Nuclear Inclusion protein a (NIa-Pro), Nuclear Inclusion protein b (NIb), and Coat protein (CP). CP is highly conserved, and its structure affects the ability for virus assembly and aphid transmission [5,6]. VPg protein performs a variety of functions in the process of virus infection [7].

The process of viral translation from mRNA to protein in eukaryotes involves more than 12 eukaryotic translation initiation factors (eIFs). Among these, the eukaryotic translation initiation factor, 4E (eIF4E), is one of the most important and it plays an important role in viral translation and the synthesis of proteins [8]. One of the key links in the establishment of systemic infection in hosts by viruses in the *Potyviridae* family is using eIF4E of the host to complete the translation. Thus, eIF4E is an indispensable host factor for potato virus Y genome translation. In most single-stranded RNA viruses, the VPg is encoded by the virus-like cap structure which interacts with the host factor, eIF4E, to initiate the translation of the viral genome [9,10]. When eIF4E mutates, its interaction with the virus VPg is eliminated or weakened, and the infectivity of the virus is lost [11,12]. Lin et al. [13] found four predominant mutations in *eIF4E1* of tobacco-resistant sources with PVY resistance. The mutations included single base insertion, substitution, and partial or complete deletion of gene fragments. Plant disease resistance induced by *eIF4E* mutants has been observed in a variety of plants. The difference between *eIF4E* in susceptible plants and *eIF4E* in resistant plants was only the difference of a single or a few amino acids in their protein sequence, but it has led to plants resistant to multiple viruses or multiple strains of a virus.

Breeding or utilizing resistant varieties is the most economical and effective means of controlling sugarcane mosaic disease. Using plant-derived genes to conduct plant antiviral research enhances biosafety [14]. Previous studies have shown that *eIF4E*-mediated natural recessive resistance is derived from the non-synonymous mutation of the eIF4E protein. Therefore, in cultivating resistant varieties, it is necessary to explore the relationship between the mutation site of *eIF4E* and disease resistance in the host. However, in sugarcane, different varieties show different levels of resistance to sugarcane mosaic disease in the field. There is no report of resistance caused by the sugarcane resistance gene, or pathogenicity change caused by the natural variation of the virus. The polymorphism of the *eIF4E* coding region in sugarcane is unknown. There are two sugarcane varieties with significant differences in resistance against sugarcane mosaic disease in the field. They are used as samples in this study. Sequence analysis was performed for both the pathogenic virus and translation initiation factor eIF4E of the host. The aim was to determine whether the difference in resistance was caused by the differences in *eIF4E* sequences between the sugarcane varieties. The non-synonymous mutation sites of the eIF4E amino acid in susceptible and resistant sugarcane varieties were explored by cloning the coding sequences of *eIF4E* in 11 different resistant materials and analyzing the polymorphisms in *eIF4E* in resistant and susceptible sugarcane varieties. The results provide guidance for the molecular breeding of sugarcane resistance to mosaic disease with *eIF4E* as the target.

## 2. Results

### 2.1. Differences in Resistance of Sugarcane Varieties against Sugarcane Mosaic Disease

Many years of field investigation of naturally occurring sugarcane diseases have shown that four resistant varieties, including CP94-1100, and seven susceptible varieties, including ROC22, have significant and stable resistance against sugarcane mosaic disease (Table 1). Of these, the incidence of ROC22 mosaic disease in different sugarcane-growing areas in Yunnan (China) reached 100%, and the whole plant had serious mosaic infection with a coverage area of 100%. The CP94-1100 sugarcane grew normally, the whole plant was relatively strong, and the leaves were clean, having no mosaic symptoms (Figure 1). CP94-1100 and ROC22 were selected as representatives of different resistant varieties, for the study of the source of resistance.

Highly resistant to sugarcane mosaic disease (R) and highly susceptible to sugarcane mosaic disease (S).

### 2.2. Detection and Sequence Analysis of SCSMV-CP and SCSMV-VPg in Different Resistant Sugarcane Varieties

*SCSMV* was found in different resistant sugarcane varieties (CP94-1100 and ROC22) through the detection of the *SCSMV-CP* and *SCSMV-VPg*, indicating that both varieties were infected by *SCSMV*. Two *SCSMV-CP* sequences obtained using RT-PCR were submitted to the GenBank database with the accession numbers ON505212 (CP94-1100) and ON505213 (ROC22). The sequence length was 938 bp, and the homologies with the published *SCSMV-CP* sequence (accession number JF488065) were 99.9% and 99.6%, respectively. The nucleotide sequence and coding proteins were basically the same. The *SCSMV-VPg* sequences were submitted to the GenBank database with the accession numbers MZ605863 and MZ605884 for CP94-1100 and ROC22, respectively. The sequence length of *SCSMV-VPg* was 594 bp, and the homologies were 99.7% and 99.0% with the published gene sequence of *SCSMV-VPg* (accession number JF488065). The nucleotide sequence and coding protein were basically the same (Table 2), indicating that the *SCSMV* isolates of the two varieties were the same. The differences in the symptoms in sugarcane were attributed to the differences in sugarcane varieties.

The standard curve for detecting SCSMV virus accumulation was Y = 2.3421X + 23.155, R^2^ ≥ 0.998, E ≥ 98.8%. It met the requirements of an ideal standard curve (R^2^ ≥ 0.98, 85% ≤ E ≤ 110) [15]. RT-qPCR showed the accumulation of SCSMV in susceptible varieties was greater than that in resistant varieties. The accumulation of SCSMV was lowest in CP94-1100 and highest in ROC22 (Figure 2).

### 2.3. Cloning and Sequence Analysis of eIF4E from Different Varieties

#### 2.3.1. Detection of eIF4E from Different Varieties

The *eIF4E* sequences of two sugarcane varieties (CP94-1100 and ROC22) were cloned, sequenced, and blasted. The homology with the coding region sequence of the published *eIF4E* (accession number KX757017) was 99.4% and 99.2%. The obtained sequence was confirmed as the coding region sequence of sugarcane *eIF4E*, named *SceIF4E-R* and *SceIF4E-S* and submitted to the GenBank database, with the accession numbers MT680884 (CP94-1100) and MT680894 (ROC22). The nucleotide and amino acid sequences of the *eIF4E* in CP94-1100 and ROC22 were significantly different, and the nucleotide sequence in the coding region was 663 bp, with a consistency of 99.1%. The amino acid sequence was 220 aa, and the consistency was 98.2% (Table 2). This result indicated that the amino acid properties of eIF4E varied greatly in the two sugarcane varieties with different levels of resistance to *SCSMV*. These results show that the difference in resistance was derived from the variation in *eIF4E*.

The standard curve for detecting eIF4E accumulation was Y = 2.2802X + 24.615, R^2^ ≥ 0.994, E ≥ 1.03%. It met the requirements of an ideal standard curve (R^2^ ≥ 0.98, 85% ≤ E ≤ 110) [15]. RT-qPCR showed the accumulation of eIF4E in susceptible varieties was greater than that in resistant varieties. The accumulation of eIF4E was the least in CP94-1100 and the most in ROC22 (Figure 2).

#### 2.3.2. Bioinformatic Analysis of Coding Region Sequence of eIF4E in Different Varieties

Bioinformatic analysis of the *eIF4E* sequences of two sugarcane varieties with different levels of resistance (CP94-1100 and ROC22) showed that there were base differences between the two varieties at 60 bp, 180 bp, 328 bp, 383 bp, 496 bp, and 661 bp in the ORF region. Of these, four sites, 60 bp, 180 bp, 328 bp, and 496 bp, led to the missense mutation of amino acids (Figure 3). These results indicate that the protein character differed between the varieties. The physicochemical properties of the proteins encoded by *SceIF4E-R* and *SceIF4E-S* were predicted online. The results show that the molecular formulas of SceIF4E-R and SceIF4E-S were C_1093_H_1652_N_306_O_328_S_7_ and C_1092_H_1649_N_307_O_330_S_7_, respectively. The relative molecular weights were 24.55 kDa and 24.58 kDa, respectively. The theoretical isoelectric points were 5.70 for both, and the instability coefficients were 29.95 and 32.29, respectively. The total average hydrophilicity values were −0.617 and −0.662, respectively, indicating that both proteins were stable and hydrophilic. The SceIF4E-R and SceIF4E-S proteins were both present in the cytoplasm, which had no transmembrane structure and signal peptide. These results indicate that SceIF4E is neither a membrane protein nor a secretory protein (Figure 4). The results of protein secondary and tertiary structure prediction show that SueIF4E-R was composed of a random coil (53.18%), alpha-helix (28.18%), extension strand (14.55%), and bate turn (4.09%), while SceIF4E-S was composed of a random coil (50.45%), alpha-helix (28.64%), extension strand (16.36%), and bate turn (4.55%) (Figure 5). Conservative domain analysis showed that both SceIF4E-R and SceIF4E-S had an IF4E conserved domain, belonging to the IF4E family of proteins; however, SceIF4E-R also contained a CDC 33 or PTZ00040 domain (Figure 6). These domains suggested that SceIF4E-R and SceIF4E-S proteins had the same function as the IF4E family of proteins, but the functions of SceIF4E-R and SceIF4E-S proteins differed.

To further clarify the functions of *SceIF4E-R* and *SceIF4E-S* and understand their phylogenetic relationships with eIF4E genes in other plants, the amino acid sequences of *eIF4E* in 10 plants were downloaded from the NCBI database (https://www.ncbi.nlm.nih.gov/, accessed on 11 June 2022) for analysis and construction of the phylogenetic tree. The 10 plants were sugarcane (*Saccharum hybrid* cultivars), sorghum (*Sorghum bicolor*), maize (*Zea mays*), Chenopodium album *(Panicum miliaceum*), millet (*Setaria italica*), *Saccharum spontaneum*, rice (*Oryza sativa*), barley (*Hordeum vulgare*), pineapple (*Ananas comosus*), and sweet potato (*Ipomoea triloba*). The phylogenetic tree was divided into three groups. The first group contained only pineapple *eIF4E*. The second group included *SceIF4E-R, SceIF4E-S,* and the *eIF4E* of eight *Gramineae* plants. The third group contained sweet potato *eIF4E* (Figure 7). The phylogenetic tree showed that all *SceIF4E* clustered in the same branch, including the *eIF4E* of sugarcane (*S. hybrid* cultivar, ASU92319) and *S. spontaneum* (AWA44706), and *SceIF4E-R* had the closest genetic relationship with *S. spontaneum* (AWA44706).

### 2.4. Polymorphism Analysis of the Coding Region Sequence of eIF4E

#### 2.4.1. Cloning of the Coding Region Sequence of eIF4E in Different Resistant Varieties

The coding region of *eIF4E* from eleven sugarcane materials comprising four resistant varieties, including CP94-1100, and seven susceptible varieties, including ROC22, was amplified using RT-PCR. Electrophoresis showed that the main bands of the PCR amplification products of all samples were clear, with no obvious heterozygous band. The size of the bands was about 660 bp, which was consistent with the expected size (Figure 8). BLAST results showed that the highest homology with sugarcane *eIF4E1* (KX757017) was 99.4%. The sequence similarity with sorghum *eIF4E1* (XM_002456973) was 97.6% and that with maize *eIF4E1* (NM_001366957) was 94.9%, indicating that the coding region sequence of sugarcane *eIF4E* was successfully obtained. It was submitted to GenBank with the accession number MT680884-MT680894.

#### 2.4.2. Nucleotide Sequence Polymorphism of the Coding Region of eIF4E

The *eIF4E* coding region sequences of 11 sugarcane materials were analyzed with respect to the open reading frame (ORF) length, insertion–deletion (InDel), single nucleotide polymorphism (SNP), nucleotide diversity index (Pi), haplotype number, and diversity (Table 3). The ORF length of 11 materials was 663 bp, forming 11 haplotypes. Each haplotype contained one material, and the haplotype diversity was 1.000. There were 28 polymorphic loci, which were SNP loci with simplified information, and no InDel locus was found. The frequency of SNP loci was 1SNP/23.7 bp. Among the twenty-eight polymorphic variation sites, eight sites were the same in the *SCSMV* materials with high resistance and high susceptibility, six variation sites only existed in high resistance to the *SCSMV* materials, and fourteen variation sites only existed in high susceptibility to *SCSMV* materials. The Pi value of all tested materials was 0.00932. The Pi value of highly resistant material (0.00930) was less than that of highly susceptible material (0.00970) (Table 3), indicating that the genetic variation of highly resistant material was less than that of highly susceptible material.

The distribution of polymorphic sites in the *eIF4E* nucleotide sequence of the tested materials is shown in Figure 9. There were multiple peaks in the previous sequence, indicating that there were many regions where the sequence changed and there may be multiple hot mutation sites. The peak appears in the region of about 400 bp, indicating that the region around 400 bp was the enrichment region of nucleotide change.

#### 2.4.3. Amino Acid Sequence Analysis of eIF4E Coding Region

The amino acid sequences corresponding to the coding region of *eIF4E* were predicted using the ORF finder (http://www.nibi.nlm.nih.gov/, accessed on 12 July 2022) and the interspecies variation of amino acid sequences of 11 sugarcane materials was analyzed using the DNAMAN software. The results show that all the samples encoded 220 amino acids. The interspecies variation analysis of amino acids showed that 203 amino acids were synonymous mutations, and 17 amino acids were non-synonymous mutations. The base mutations were all substitutions, and there was no insertion or deletion. A total of 11 proteins were encoded. Eighteen base mutations in 28 SNPs resulted in 17 amino acid non-synonymous mutations, which were SNP46, SNP60, SNP71, SNP172, SNP180, SNP193, SNP261, SNP304, SNP328, SNP406, SNP415, SNP416, SNP433, SNP448, SNP451, SNP493, SNP496, and SNP501. SNP415 and SNP416 resulted in amino acid non-synonymous mutations at the same site (Table 4). Of all the amino acid non-synonymous mutation sites, only four (SNP193, SNP261, SNP406, and SNP451) occurred in highly resistant *SCSMV* materials and only ten (SNP46, SNP60, SNP180, SNP304, SNP328, SNP415, SNP433, SNP448, SNP493, and SNP501) occurred in highly susceptible *SCSMV* materials. To evaluate the natural selection pressure of the eIF4E gene, the ratio of synonymous mutation (Ks) to non-synonymous mutation (Ka) (Ka/Ks) of *eIF4E* sequences obtained from all materials was analyzed, yielding a value of 0.3727 (less than 1), indicating that the eIF4E gene was subjected to negative selection pressure as a whole.

### 2.5. Spatial Structure of eIF4E Protein

Wheat eIF4E protein (2idr.1.A) was selected as the model protein using SWISS-MODEL software to predict the spatial structures of 11 proteins encoded by *eIF4E*. The spatial structures of 11 proteins showed high similarity. Previous studies showed that the cap structure of eIF4E played an indispensable role in protein function [16,17]. The key regions of the cap structure are shown in the A and B regions of Figure 9. Region A is located in the 57–73 amino acid site range of eIF4E, and region B is located in the 107–118 amino acid site range. In this study, the amino acid variation sites, 58, 60, and 64 sites, were all located in region A. Of these, the 58 amino acid variation site was the variation site of all materials relative to the control sequence, KX757017. The 64 amino acid variation site was only found in Protein 3 encoded by the eIF4E gene of Yuetang 55, an extremely resistant material, and the amino acid variation was Ala64Val. The 60 amino acid variation site was only found in Protein 11 encoded by the eIF4E gene of ROC22, an extremely susceptible material, and the amino acid variation was Lys60Glu (Table 4 and Table 5, Figure 10).

## 3. Discussion

SCSMV is one of the main viral pathogens of sugarcane mosaic disease. Since the first detection of SCSMV in Yunnan in 2011 [18], it has spread rapidly, with strong pathogenicity, and has become widespread in all sugarcane areas in China, causing serious harm. A detection rate of 100% has revealed it as the main pathogen of sugarcane mosaic disease in China [19,20]. SCSMV is a member of the potato virus Y family, and the nucleotide sequence variation of its cp gene is highly limited [21]. The VPg protein encoded by the SCSMV genome is a key protein in plants infected by the virus [22]. Therefore, the CP and VPg proteins of the virus are essential for successfully infecting the host. In this study, the cp and vpg sequences of SCSMV in different varieties infected by SCSMV were analyzed to determine differences in the pathogenicity of SCSMV among different varieties. The results show that the nucleotide and amino acid sequences of the cp and vpg genes of SCSMV belonging to the same isolate were highly consistent (more than 99.0%) in two sugarcane varieties with different resistances. These results indicate that the differences in resistance between the varieties were not caused by the differences in SCSMV. The results of this study are consistent with those reported by Wang et al. [20] and Zhang et al. [23]. The Chinese isolates of SCSMV had obvious geographical variations in characteristics, but there was no obvious genetic differentiation within the population. Quantification of SCSMV accumulation levels in different varieties show that the accumulation of SCSMV in susceptible varieties was greater than that in resistant varieties. Furthermore, the eIF4E accumulation levels in different varieties were quantified. It was also found that the accumulation level of eIF4E in susceptible varieties was also greater than that in resistant varieties. These results indicated that eIF4E may contribute to the accumulation of SCSMV in sugarcane.

The virus has a simple structure, and lacks a cap structure that is required to start translation. After the virus enters a plant, the VPg encoded by the virus interacts with the eIF4E of the plant to complete virus replication and infection [24]. Therefore, *eIF4E* is considered to be a recessive resistance gene against viruses in many plants. The genome translation efficiency of viruses in the *Potyviridae* family (including potato virus Y and barley yellow mosaic virus) depends on the interaction level of VPg and eIF4E [25,26]. Studies have shown that the introduction of *eIF4E* can enable the replication of wheat mosaic virus (WYMV) in barley protoplasts [27]. Therefore, the resistance of different sugarcane varieties is considered to be caused by differences in the translation efficiency of the *eIF4E* sequences of the varieties. The results of this study show that the amino acid properties of *eIF4E* were quite different between the two sugarcane varieties with different levels of resistance to *SCSMV*, suggesting that the ability of eIF4E to recruit subsequent translation initiation factors and ribosome subunits differed, leading to the difference in resistance derived from the variation of *eIF4E*. Further bioinformatics analysis of the *eIF4E*-ORF of two sugarcane varieties with different levels of resistance (CP94-1100 and ROC22) showed that the physicochemical properties of the two proteins were basically the same, but there were four-base missense mutations in the ORF region, resulting in different conservative domains. Although SceIF4E-R and SceIF4E-S belonged to the IF4E family of proteins and had the same function as the IF4E family of proteins, SceIF4E-R contained CDC 33 or PTZ00040 domains, indicating that SceIF4E-R may have other functions. In phylogenetic analysis, SceIF4E-R and SceIF4E-S clustered with eight *Gramineae* crops, in different groups from pineapple and sweet potato, indicating that the eIF4E had obvious species differences and high homology within species. The results of this study were consistent with the results of Geng et al. [28] on *eIF4E* sequence analysis of wheat varieties. SceIF4E-R is closely related to *S. spontaneum*, indicating that they have similar functions. Huang et al. [29] analyzed the phenotypic genetic diversity of the agronomic traits of *S. spontaneum* resources at home and abroad. The results show that *S. spontaneum* is highly resistant to mosaic disease. Li et al. [30] pointed out that *S. spontaneum* was a promising resistant germplasm for breeding sugarcane varieties which were resistant to mosaic disease. In this study, SceIF4E-R was cloned from the resistant germplasm, CP94-1100, and was closely related to *S. spontaneum*, indicating that SceIF4E-R may have the function of resistance to mosaic disease.

Polymorphism analysis of 11 sugarcane varieties with extremely high resistance and high susceptibility to *SCSMV* showed that the coding region of *eIF4E* was 663 bp in length, with 28 SNPs and no InDel locus. The SNP frequency was 1 SNP/23.7 bp, slightly lower than that of the sugarcane tillering key gene, *HTD2* [31], and higher than that of the tropical sucrose synthase gene *SuSy* (1SNP/108 bp) [32]. The SNP frequency of 1 SNP/23.7 bp was much higher than the polymorphism of cabbage (1 SNP/150 bp) [33] and soybean (1 SNP/272 bp) [34]. SNP is a gene sequence polymorphism at the genome level caused by a single nucleotide variation in a closely related population or conspecific individual. It is caused by single base conversion, transversion, and single base insertion, and deletion. SNP located in the non-coding region does not affect the amino acid sequence of its protein, and SNP located in the gene coding region will change the amino acid sequence of its protein, thus changing the function of the protein [35]. To sum up, SNP frequency is related to plant species, gene types, and functions.

Synonymous and non-synonymous mutations in the coding region of genes reflect variation in genes, and the ratio of their mutation rates (the Ka/Ks value) reflect the effect and direction of gene selection. A Ka/Ks value of more than 1 indicates that the gene is affected by positive selection and belongs to the rapid evolution gene; Ka/Ks = 1 indicates that synonymous mutation and non-synonymous mutation frequency are the same and the population is not affected by selection pressure; a Ka/Ks value less than 1 indicates that the gene is affected by negative selection and is a relatively conservative gene [36]. In this study, the Ka/Ks value of the coding region of *eIF4E* was less than 1, indicating that the eIF4E gene was affected by negative selection and was a relatively conservative gene, consistent with the results of Ruffel et al. [37] on the conservation of the eIF4E amino acid sequence in higher plants.

In plants, cloning and utilizing natural resistance genes have important applications in the cultivation of disease-resistant varieties. The absence or mutation of *eIF4E* in host plants can lead to recessive resistance and hinder infection by viruses in the family *Potyviridae*. Studies have shown that the natural recessive resistance mediated by *eIF4E* is derived from the non-synonymous mutation of the eIF4E protein [38,39]. Stein et al. [40] conducted a sequence analysis on *eIF4E* in *Barley Yellow Mosaic Virus*-resistant barley and found that the mutated amino acid residues, namely, Ser57, Lysl18, Thrl20, Asnl60, Glnl61, Ser205, Asp206, and Gly208, were located on both sides of the cap-binding pocket of the elF4E. The analysis of a three-dimensional structure model of the Arabidopsis eIF (iso) 4E protein revealed that mutant amino acid residues (G1y43, Lysl02, Thrl04, Lys149, Glnl50, Ser194, Asp195, and Thr197) corresponding to barley were found near the cap-binding pocket. A related analysis showed that the tryptophan at positions 46 and 92 in Arabidopsis eIF (iso) 4E was located in the hat recognition region. When these two tryptophans were mutated to leucine, eIF (iso) 4E lost its ability to bind to virus VPg and methylguanosine [41]. Related studies showed that the combination of VPg and eIF4E occurred near the cap pocket [16,17]. These indicate the mutation sites of *eIF4E* were primarily concentrated near the cap pocket. In this study, 11 tested materials were divided into 11 haplotypes, encoding 11 proteins. A total of 28 SNPs were detected in the coding region, of which 18 SNP loci could cause 17 amino acid sites with non-synonymous mutations. The three-dimensional structure analysis of the protein showed that the two amino acid non-synonymous mutation sites were located near the cap pocket. The nucleotides at the 193 bp site of Yuetang 55 were mutated from C to G, and the encoded 64-position alanine was mutated to valine. The nucleotide at the 180 bp site of ROC22 was mutated from A to G, and the encoded 60-lysine was mutated to glutamic acid. It is these two sites that may be the resistance sites of the eIF4E gene to *SCSMV* or *SCSMV* sensitivity in sugarcane. The next step is to study whether the protein formed by the mutation in these two sites can bind to VPg and develop the marker to marker-assisted selection for breeding *SCSMV*-resistant cultivars. 

## 4. Materials and Methods

### 4.1. Tested Material

Eleven varieties with stable and universal resistance against sugarcane mosaic disease in the field [42] served as samples. Susceptible varieties were denoted S, while resistant samples were denoted R (Table 1). Healthy leaves of the resistant varieties (without mosaic symptoms) were collected, and the mosaic leaves of susceptible varieties (with typical mosaic symptoms) were stored at −80 °C.

### 4.2. Design and Synthesis of Primers

SCSMV-CP-F/R primers and SCSMV-qCP-F/R were designed to amplify the *cp* sequence of *SCSMV* according to the whole genome sequence of SCSMV-JP2 (accession number JF488065) reported in the GenBank database. SCSMV-VPg-4F/4R primers were designed to amplify the *vpg* sequence and partial Nia-pro nucleotide sequence of SCSMV-GN12 (accession number KT257273) reported in the GenBank database. eIF4E-5F/5R primers and eIF4E-qF/qR were designed according to the reported sequence of the sugarcane translation initiation factor, *eIF4E* (accession number KX757017), in the GenBank to amplify the *eIF4E* sequence. The primer sequences (Table 6) were synthesized by Shanghai Sangon Biotechnology Co., Ltd. (Shanghai, China).

### 4.3. Extraction of Total RNA from Plants

Total RNA was extracted from 0.2 g of tested material using a TransZol Plant kit (TransGen Biotech, Co., Ltd., Beijing, China). The specific method was implemented according to the kit instructions, and the RNA precipitation was dissolved in sterile water treated with 30 μL of DEPC. The total RNA was submitted as a template to synthesize cDNA using TransScript One-Step gDNA Removal and cDNA Synthesis SuperMix kit (TransGen Biotech Co., Ltd.) according to the instructions.

### 4.4. Gene Cloning

cDNA was used as the template, and PCR amplification was carried out using primers SCSMV-CP-F/R, SCSMV-VPg-4F/4R, and eIF4E-5F/5R. PCR amplification of *SCSMV-CP* was performed in a 25 µL reaction mixture, including 9.5 µL of ddH_2_O, 12.5 µL of 2 × Easy Taq PCR SuperMix (TransGen Biotech Co., Ltd., Beijing, China), 2.0 µL of cDNA template, and 0.5 µL of upstream and downstream primers each (20 µg/µL). In the PCR amplification of *SCSMV-VPg* and *eIF4E*, the cDNA template and upstream and downstream primers were 1.0 µL. The thermal cycling conditions of *SCSMV-CP* were as follows: 5 min at 94 °C followed by 35 cycles for 30 s at 94 °C, 30 s at 50 °C, and 1 min at 72 °C, with a final extension for 10 min at 72 °C. The thermal cycling conditions of *SCSMV-VPg* and *eIF4E* were as follows: 5 min at 94 °C followed by 35 cycles for 30 s at 94 °C, 30 s at 60 °C, and 1 min at 72 °C, with a final extension for 10 min at 72 °C. The amplified product (10 µL) was analyzed through electrophoresis on 1% agarose gel stained with ethidium bromide. The PCR products were purified according to the manufacturer’s instructions on the DNA agarose gel purification kit (Tiangen Biotech Co., Ltd., Beijing, China). The purified PCR products were cloned with pEASY-T5 Zero Cloning Kit (TransGen Biotech Co., Ltd., Beijing, China) and transformed into *Escherichia coli* cells Trans1-T1 (TransGen Biotech Co., Ltd., Beijing, China). Six positive clones per sample were sequenced. 

### 4.5. SCSMV Accumulation Detection

The positive CP gene plasmid with correct sequencing was extracted. The plasmid DNA concentration was determined by nucleic acid protein analyzer. The copies of plasmid concentration were calculated according to the follow formula [43]. Then, the plasmid was used as a standard sample. The ChamQ Universal SYBR qPCR Master Mix kit (Vazyme, Nanjing Biological Company) was used to perform qPCR with primers SCSMV-qCP-F/R or eIF4E-qF/qR. The standard sample was successively diluted into 5 plasmid samples with 10-times gradient, and then to make the standard curve by qPCR. The DNA concentration of all samples was uniformly quantified to 500 ng/μL as template DNA for qPCR. The obtained Ct values were used to calculate the copies of SCSMV according to regression equation of the standard curve.
Copies (copies·µL^−1^) = [6.02 × 10^23^ (copies·mol^−1^) × DNA concentration (ng·µL^−1^) × 10^−9^]/[base number (bp) × 660 (ng·mol^−1^)]

### 4.6. Sequence Analysis

To identify homology, the sequences were compared with the published sequences in GenBank using the BLAST function on the NCBI website. DNAMAN software was used for sequence multiple alignments. The nucleotide sequence and amino acid sequence differences between the samples and the control KX757017 were analyzed. Single nucleotide polymorphism (SNP) and insertion–deletion (Indel) sites were analyzed. The open reading frame (ORF) of *eIF4E* was searched using the online tool, ORF Finder, and the amino acid sequence was deduced. The conserved domain of the eIF4E protein was analyzed using the online tool, Conserved Domain Search. ProtScale (https://web.expasy.org/protscale/, accessed on 25 May 2022) and ProtParam (https://web.expasy.org/, accessed on 25 May 2022) in ExPASy server ProtParam/) were used to analyze the amino acid residues, theoretical molecular weight, isoelectric point, instability index, hydrophilicity, and hydrophobicity of the encoded protein. The signal peptide and transmembrane region of amino acid sequence were analyzed using SignalP 5.0 Server (https://services.healthtech.dtu.dk/service.php/SignalP-5.0, accessed on 26 May 2022) and TMHMM Server V2.0 (https://services.healthtech.dtu.dk/service.php/TMHMM-2.0, accessed on 27 May 2022). Protein secondary structure was predicted using SOPMA (https://npsa-prabi.ibcp.fr/cgi-bin/npsa_automat.pl/page=npsa_sopma.html, accessed on 1 July 2022), and the tertiary structure was predicted using SWISS-MODEL (https://swissmodel.expasy.org/interactive, accessed on 16 July 2022). Subcellular localization was predicted using Plant-mPLoc (http://www.csbio.sjtu.edu.cn/cgi-bin/PlantmPLoc.cgi, accessed on 10 June 2022). The eIF4E amino acid sequences of other species were downloaded from the NCBI database, and the sequence analysis and multiple sequence alignment were performed using DNAMAN and ClustalW software. The phylogenetic tree was constructed using the neighbor-joining method and Kimura’s two-parameter model as implemented in the MEGA version 6.0 [44]. The bootstrap value was 1000 replicates. Haplotype diversity (Hd), nucleotide diversity (Pi), standard deviation (SD), number of polymorphic loci (S), and Ka/Ks were calculated using the Dnasp v5.1 software.

## Figures and Tables

**Figure 1 plants-12-02805-f001:**
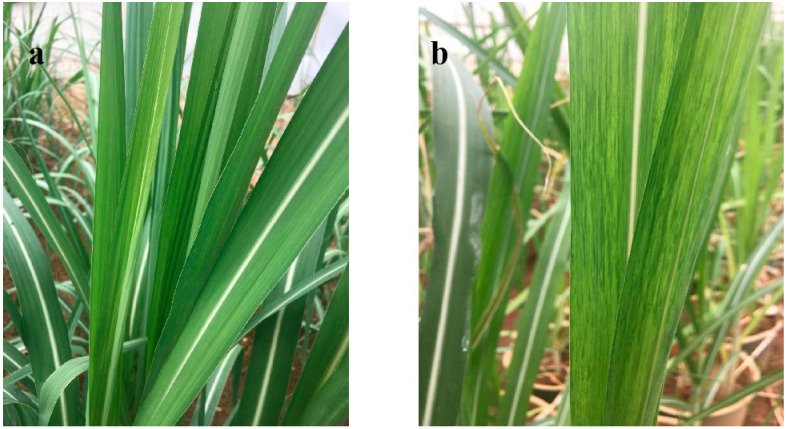
Symptoms of sugarcane mosaic disease in different varieties of sugarcane in Yunnan Province, China. From left to right, varieties are CP94-1100 (**a**), and ROC22 (**b**).

**Figure 2 plants-12-02805-f002:**
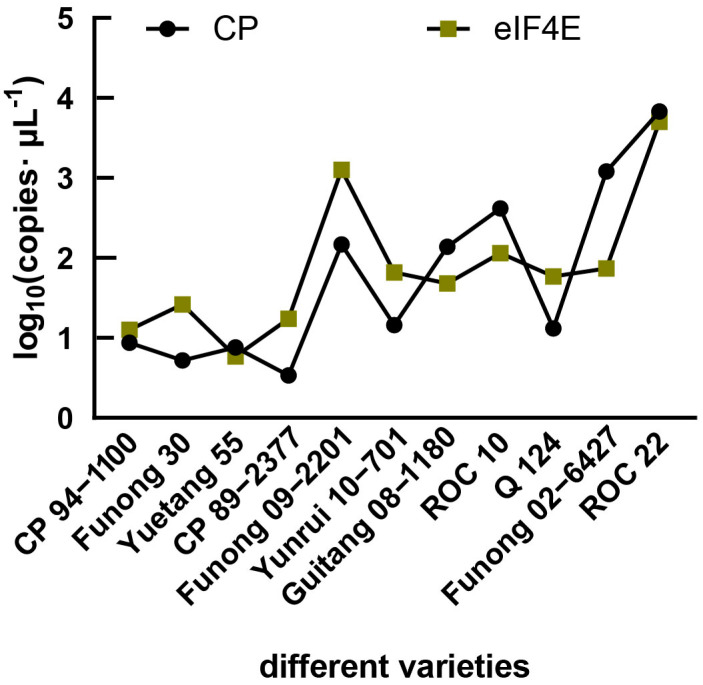
Quantification of accumulation levels for SCSMV and eIF4E.

**Figure 3 plants-12-02805-f003:**
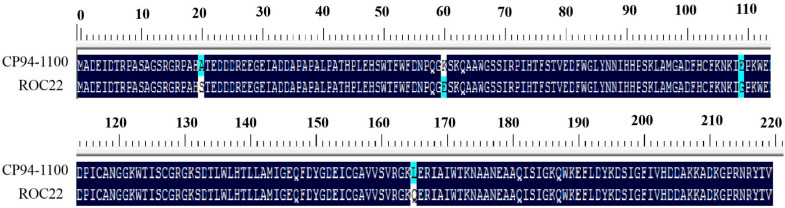
Alignment analysis of amino acid sequences of eIF4E in sugarcane.

**Figure 4 plants-12-02805-f004:**
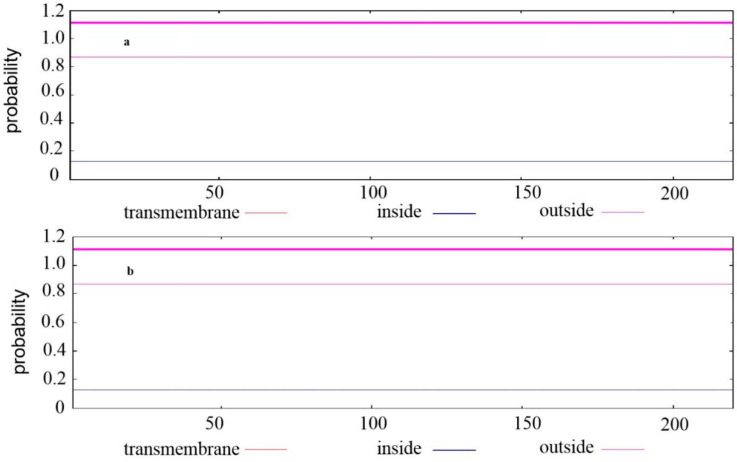
Prediction and analysis of transmembrane domains of SceIF4E. SceIF4E-R (**a**) and SceIF4E-S (**b**).

**Figure 5 plants-12-02805-f005:**
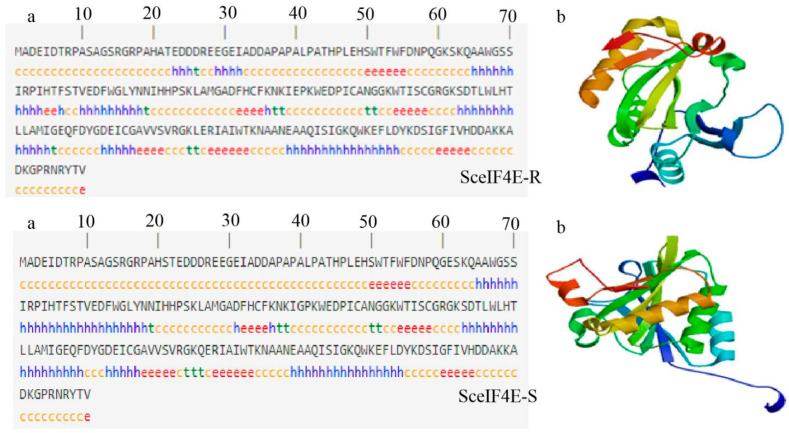
Protein secondary and tertiary structures of SceIF4E. Secondary structure (**a**): orange c represents random coil, red e represents extended strand, blue h represents alpha helix, and green t represents beta turn; and tertiary structure (**b**). Blue h: alpha helix; red e: extended strand; orange c: random coil; green t: bate turn.

**Figure 6 plants-12-02805-f006:**
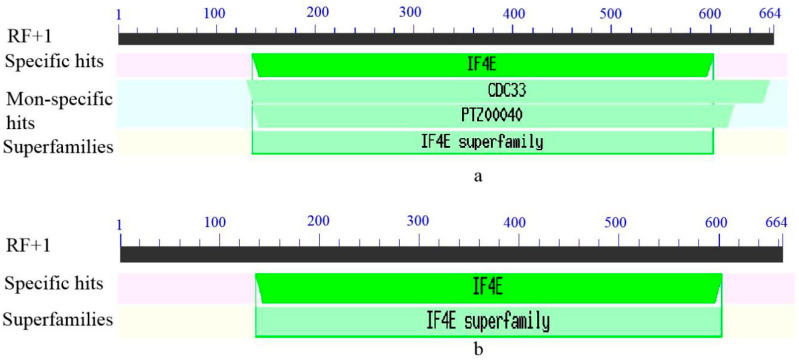
Conserved domain of SceIF4E. SceIF4E-R (**a**) and SceIF4E-S (**b**).

**Figure 7 plants-12-02805-f007:**
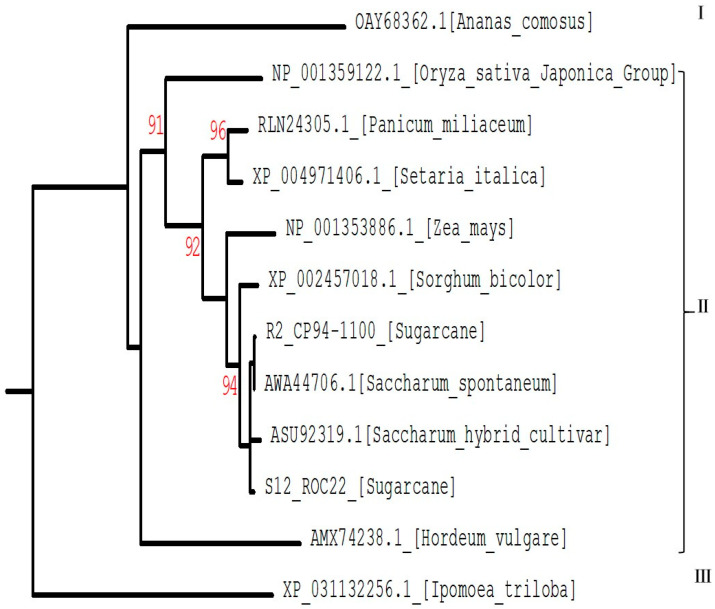
Phylogenetic tree of eIF4E protein in sugarcane and ten plants. Red numbers on branches indicate bootstrap values based on 1000 replicates (values < 90 are not shown).

**Figure 8 plants-12-02805-f008:**
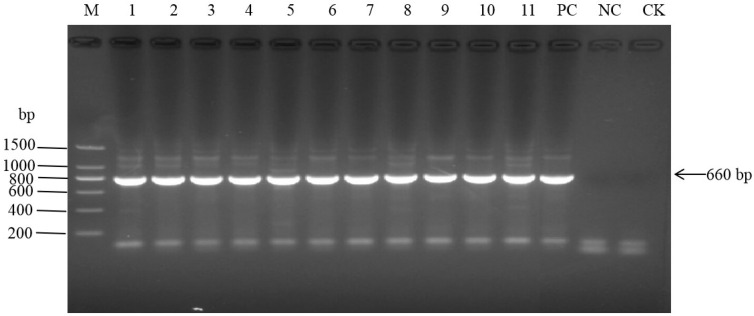
Detection of *eIF4E1* by PCR in different sugarcane varieties. From left to right, marker (M), samples from Nos. 1–11 (Lanes 1–11), positive control (PC), negative control (NC), blank control (CK).

**Figure 9 plants-12-02805-f009:**
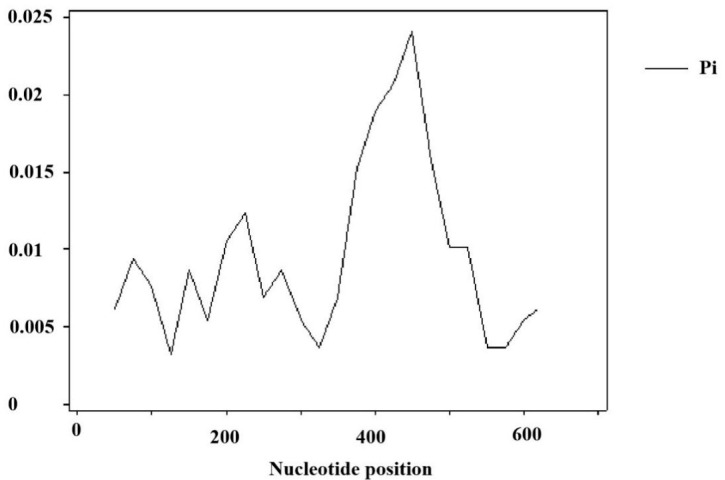
Distribution of the polymorphic sites of *eIF4E* in sugarcane.

**Figure 10 plants-12-02805-f010:**
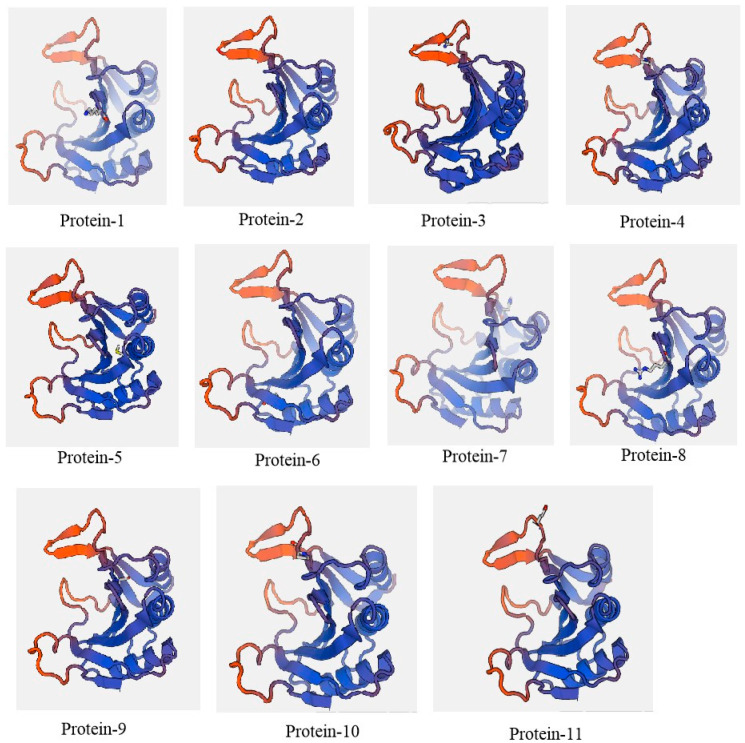
Protein 3D structure and mutation sites of *eIF4E*-coding proteins.

**Table 1 plants-12-02805-t001:** Samples and phenotype of resistance for *sugarcane streak mosaic virus* in sugarcane.

Number	Sample	Phenotype	Number	Sample	Phenotype
1	CP 94-1100	R	7	Guitang 08-1180	S
2	Funong 30	R	8	ROC 10	S
3	Yuetang 55	R	9	Q 124	S
4	CP 89-2377	R	10	Funong 02-6427	S
5	Funong 09-2201	S	11	ROC 22	S
6	Yunrui 10-701	S			

**Table 2 plants-12-02805-t002:** Sequence analysis of SCSMV and eIF4E for sugarcane CP94-1100 and ROC22.

Varieties	CP		*VPg*		eIF4E	
	Nucleotide	Amino Acid	Nucleotide	Amino Acid	Nucleotide	Amino Acid
*ROC22*	938	240	594	198	663	220
CP94-1100	938	*240*	*594*	*198*	*663*	*220*
*Sequence identities/%*	99.5	*99.2*	*98.2*	*99.0*	*99.1*	*98.2*

Note: *Sequence identities*—The homologies of the varieties ROC22 and CP94-1100 on nucleotide sequence and amino acid sequence, respectively.

**Table 3 plants-12-02805-t003:** Analysis of polymorphism of *eIF4E* cDNA in sugarcane.

	Length	No. of Haplotypes	Haplotype Diversity (Hd ± SD)	Nucleotide Diversity (Pi ± SD)	No. of Polymorphic Sites (S)	Specific Polymorphic Sites
All samples	663	11	1.000 ± 0.039	0.00932 ± 0.0011	28	SNP3; SNP101; SNP172; SNP242; SNP383; SNP416; SNP419; SNP496
Highly resistant to SCSMV	663	4	1.000 ± 0.177	0.00930 ± 0.0020	14	SNP181; SNP193; SNP261; SNP406; SNP451; SNP614
Highly susceptible to SCSMV	663	7	1.000 ± 0.076	0.00970 ± 0.00153	22	SNP46; SNP60; SNP180; SNP304; SNP328; SNP415; SNP433; SNP437; SNP448; SNP493; SNP501; SNP554; SNP644; SNP662

**Table 4 plants-12-02805-t004:** The types of SNP mutation of *eIF4E* gene in sugarcane.

Number	Mutation	Polymorphic Type	Amino Acid Variations	Mutation Type	Material
1	SNP46	G/A	GLY15Asp	Non-synonymous mutation	Hap7
2	SNP60	G/T	Ala20Ser	Non-synonymous mutation	Hap7; Hap10; Hap11
3	SNP71	G/C	Glu23Asp	Non-synonymous mutation	Hap1; Hap2; Hap3; Hap4; Hap5; Hap6; Hap7; Hap8; Hap9; Hap10; Hap11
4	SNP101	T/C	Asp33Asp	Synonymous mutation	Hap3; Hap8
5	SNP172	A/C	Gln58Pro	Non-synonymous mutation	Hap1; Hap2; Hap3; Hap4; Hap5; Hap6; Hap7; Hap8; Hap9; Hap10; Hap11
6	SNP180	A/G	Lys60Glu	Non-synonymous mutation	Hap11
7	SNP181	A/G	Ser61Ser	Synonymous mutation	Hap2
8	SNP193	C/G	Ala64Val	Non-synonymous mutation	Hap3
9	SNP242	G/A	Glu80Glu	Synonymous mutation	Hap2; Hap4; Hap7; Hap9
10	SNP261	A/G	Asn87Asp	Non-synonymous mutation	Hap2
11	SNP304	T/C	Phe101Ser	Non-synonymous mutation	Hap9
12	SNP328	A/G	Glu109Gly	Non-synonymous mutation	Hap11
13	SNP383	T/C	Cys127Cys	Synonymous mutation	Hap1; Hap7
14	SNP406	T/C	Leu135Pro	Non-synonymous mutation	Hap2
15	SNP415	A/G	His138Arg	Non-synonymous mutation	Hap5
16	SNP416	C/T	His138Arg		Hap4; Hap9
17	SNP419	T/C	Th139Thr	Synonymous mutation	Hap3; Hap5
18	SNP433	T/C	Ile144Thr	Non-synonymous mutation	Hap5
19	SNP437	C/T	Gly145Gly	Synonymous mutation	Hap5
20	SNP448	A/G	Asp149Gly	Non-synonymous mutation	Hap8
21	SNP451	A/G	Tye150Cys	Non-synonymous mutation	Hap2
22	SNP493	A/G	Lys164Arg	Non-synonymous mutation	Hap7
23	SNP496	A/T A/G	Gln165Leu Gln165Arg	Non-synonymous mutation	Hap1; Hap7; Hap8
24	SNP501	A/G	Arg167Gly	Non-synonymous mutation	Hap6
25	SNP554	T/C	Ile184Ile	Synonymous mutation	Hap8
26	SNP614	C/T	Asp204Asp	Synonymous mutation	Hap3
27	SNP644	G/A	Arg214Arg	Synonymous mutation	Hap9
28	SNP662	G/A	Val220Val	Synonymous mutation	Hap11

**Table 5 plants-12-02805-t005:** Amino acid variation of eIF4E protein sequences between KX757017 and sugarcane used in this study.

Varieties	Haplotype	Protein	Amino Acid Variation Sites																
			15	20	23	58	60	64	87	101	109	135	138	144	149	150	164	165	167
KX757017			G	A	E	Q	K	A	N	F	E	L	H	L	D	Y	K	Q	R
CP 94-1100	Hap1	Protein-1	—	—	D	P	—	—	—	—	—	—	—	—	—	—	—	L	—
Funong 30	Hap2	Protein-2	—	—	D	P	—	—	D	—	—	P	—	—	—	C	—	—	—
Yuetang 55	Hap3	Protein-3	—	—	D	P	—	V	—	—	—	—	—	—	—	—	—	—	—
CP 89-2377	Hap4	Protein-4	—	—	D	P	—	—	—	—	—	—		—	—	—	—	—	—
Funong 09-2201	Hap5	Protein-5	—	—	D	P	—	—	—	—	—	—	R	T	—	—	—	—	—
Yunrui 10-701	Hap6	Protein-6	—	S	D	P	—	—	—	—	—	—	—	—	—	—	—	—	G
Guitang 08-1180	Hap7	Protein-7	D	—	D	P	—	—	—	—	—	—	—	—	—	—	R	L	—
ROC 10	Hap8	Protein-8	—	—	D	P	—	—	—	—	—	—	—	—	G	—	—	R	—
Q 124	Hap9	Protein-9	—	—	D	P	—	—	—	S	—	—	—	—	—	—	—	—	—
Funong 02-6427	Hap10	Protein-10	—	S	D	P	—	—	—	—	—	—	—	—	—	—	—	—	—
ROC 22	Hap11	Protein-11	—	S	D	P	E	—	—	—	G	—	—	—	—	—	—	—	—

The same amino acids with KX75701 (—).

**Table 6 plants-12-02805-t006:** Primers used in this study.

Primers	Sequence (5′ to 3′)	Product Size	Target Gene
SCSMV-CP-F	ACAAGGAACGCAGCCACCT	938 bp	*CP*
SCSMV-CP-R	ACTAAGCGGTCAGGCAAC		
SCSMV-VPg-4F	GGGAAGAAGCGTCGAACTCA	712 bp	*vpg* + Partial NIa-Pro
SCSMV-VPg-4R	CAACACACCAACTCTGCGTG		
eIF4E-5F	ATGGCCGACGAGATCGACAC	660 bp	*eIF4E*
eIF4E-5R	GCAACTCCTCGGCAAATACAG		
SCSMV-qCP-F	AACAACAACGAGTCAAGCTG	143 bp	SCSMV
SCSMV-qCP-R	AGAGATGAGAGCTTGTGGTG		
eIF4E-qF	GGCGAACAATTCGACTATGG	93 bp	*eIF4E*
eIF4E-qR	TCTGAGCAGCTTCATTAGCA		

## Data Availability

All authors agree with MDPI Research Data Policies.

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
