# Peer review of "Relationship between Sugarcane eIF4E Gene and Resistance against Sugarcane Streak Mosaic Virus"

_plants, 2023, doi:10.3390/plants12152805_

Round 1

Reviewer 1 Report

I think the work presented in this manuscript is very interesting from the point of view of studying the molecular variability of eIF4E in diverse sugarcane varieties. However, I  think that the title of the paper does not adequately describe the outcomes presented here since I do not see that the results presented in this manuscript prove that differences in resistance to SCSMV are due to the inherent differences in eIF4E of the sugarcane varieties.

In this paper the authors the authors continually refer to the concepts of resistant and susceptible sugarcane varieties. In fact, CP94-1100 and ROC22 varieties were selected as representatives of different resistant varieties against the virus. As far as I understand, eiF4E correspond to a recessive resistance family that prevents infection by some viruses (mainly potyviruses) and, although subliminal multiplication within cells with inactivated eIF4E isoforms have been reported, spread in plant tissues are restricted. So, depending on the eIF4E isoforms, the plant is resistant or susceptible to a specific virus.

Furthermore, the authors found that SCSMV accumulated in both resistant CP94-1100 and susceptible ROC22 varieties. I can be totally wrong, but I understood from the work referenced as #41 (DOI 10.1007/s10658-017-1147-3) that in that study leaves showing mosaic symptoms were collected from a large number of varieties, including Funong 30 and Yuetang 55, and the presence of SCSMV was confirmed.

Main concerns:

1-What the authors understand as highly resistant and highly susceptible plants to SCSMV?

2- Are there degrees of resistance regarding eiF4E related susceptibility? How that is measured? According to the symptom’s appearance? According to the viral accumulation levels?

3-Are there differences in the SCSMV accumulation levels in the resistant and susceptible varieties?. Quantification of viral accumulation levels should be presented

  One of the observations on which this work is based is that, since there are no changes in the VPg of the virus infecting both varieties, the resistance or susceptibility of those sugarcane varieties must necessarily fall on differences on eiF4E.   Main concern: 4- To support this premise, it should be experimentally demonstrated that expression of a susceptible eiF4E isoform in the resistant CP94-1100 variety would break the resistance to the virus. I do not know if transient or stable expression of a foreign protein is could be carried out in sugarcane

Minor concerns:

I am not a native English but I found several sentences scattered along the manuscript which I find confusing. Some examples:

“Compared with susceptible eIF4E, resistant eIF4E encode only one or a few amino acids in the protein sequence”.   “Therefore, in this study, two sugarcane varieties with significant differences in resistance against sugarcane mosaic disease in the field were used as samples and sequences were analysed in the translation initiation factor, eIF4E, of the host and the pathogen virus”.

Reviewer 2 Report

Dear autors,

I had a pleasure to read a well-written manuscript entitled Identification of eIF4E as the host factor for Sugarcane streak mosaic virus resistance. The topic is intriguing and compelling.

However, my main concern is that the polimerase used is recommended, according to the manufacturer’s instructions,  for routine PCR, High throughput PCR and colony PCR. It is not an high fidelity polimerase. In a paper like this where the results are based on the sequencing of genes, it is more convenient to use an high fidelity Taq or another type of polimerase whose error rate is minimum and almost zero. For these reasons I cannot recommend this paper to be published.

Round 2

Reviewer 2 Report

Read the authors' answer, any other reason is recognised to prevent the publication on your manuscript.

Regards